# Satellites Reveal Mobility: A Commuting Origin-destination Flow Generator for Global Cities

**Can Rong**[1,2] *  **Xin Zhang**[1]   **Yanxin Xi**[3]   **Hongjie Sui**[1]   **Jingtao Ding**[1][†]  **Yong Li**[1][‡]

[1]Dept. Electronic Engineering, BNRist, Tsinghua University
[2]Singapore-MIT Alliance for Research and Technology
[3]Dept. Computer Science, University of Helsinki

## Abstract

Commuting Origin-destination (OD) flows, capturing daily population mobility of citizens, are vital for sustainable development across cities around the world. However, it is challenging to obtain the data due to the high cost of travel surveys and privacy concerns. Surprisingly, we find that satellite imagery, publicly available across the globe, contains rich urban semantic signals to support high-quality OD flow generation, with over 98% expressiveness of traditional multisource hard-to-collect urban sociodemographic, economics, land use, and point of interest data. This inspires us to design a novel data generator, GlODGen (**Gl**obal-scale **O**rigin-**D**estination Flow **Gen**erator), which can generate OD flow data for any cities of interest around the world. Specifically, GlODGen first leverages Vision-Language Geo-Foundation Models to extract urban semantic signals related to human mobility from satellite imagery. These features are then combined with population data to form region-level representations, which are used to generate OD flows via graph diffusion models. Extensive experiments on 4 continents and 6 representative cities show that GlODGen has great generalizability across diverse urban environments on different continents and can generate OD flow data for global cities highly consistent with real-world mobility data. We implement GlODGen as an automated tool, seamlessly integrating data acquisition and curation, urban semantic feature extraction, and OD flow generation together. It has been released at `https://github.com/tsinghua-fib-lab/generate-od-pubtools`.

## 1   Introduction

Commuting origin-destination (OD) flows profile the regular population movement in daily life between every two urban regions within a given urban area [55], providing a critical foundation for various applications. For example, OD flows are essential inputs for simulations and analyses in traffic management and urban planning, serving as the travel demands of citizens for developing more informed policies [12, 66]. Recent studies have also explored the potential of OD flows in supporting research related to the United Nations Sustainable Development Goals (SDGs) on community detection [15], urban resilience [52, 28], pubic health [37] and environmental protection [81], further demonstrating their importance. Therefore, obtaining OD flows holds great significance for cities around the world, especially as they increasingly support interdisciplinary research efforts that span traditional and emerging urban challenges [55, 6].

However, it is very challenging to obtain this valuable data. Traditional methods for obtaining OD flows, such as door-to-door travel surveys [55] and the aggregation of large-scale individual mobility

---

*Work primarily done at Tsinghua University.

[†]Corresponding author: dingjt15@tsinghua.org.cn

[‡]Corresponding author: liyong07@tsinghua.edu.cn

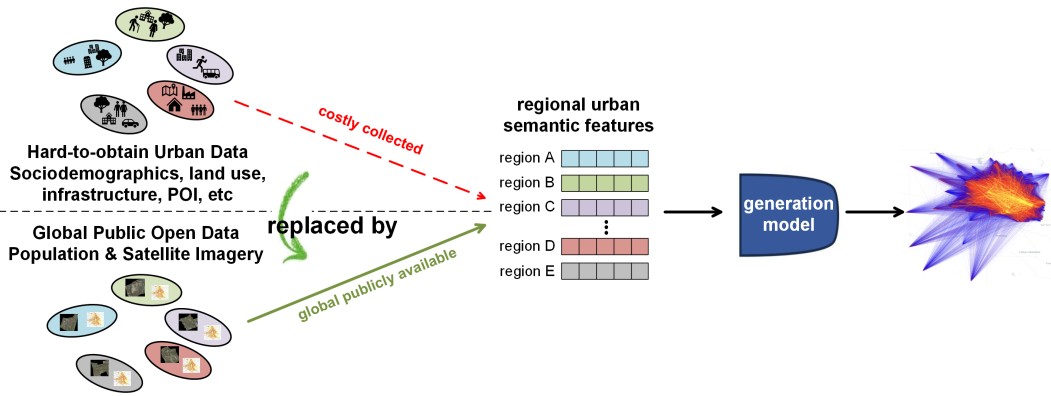

Figure 1: An illustration of replacing the traditional hard-to-collect urban data by easily accessible global public data (e.g., satellite imagery and population data) in OD flow generation.

trajectories [11, 34, 22, 49], are often infeasible in target cities due to high costs and privacy concerns. Although recent studies have explored physical [85, 64, 6] and computational [51, 53, 63, 44, 58, 57, 56] models for OD flow generation, these approaches face key limitations. Physical models rely solely on population distribution and adopt overly simplistic assumptions, ignoring urban semantic differences and resulting in limited accuracy. Computational models perform better but still rely on hard-to-obtain, expensive, and fine-grained inputs such as sociodemographics, infrastructure data, land use, and points of interest (POIs), which limits their applicability to only one or a small number of data-rich, developed cities.

Recent studies [73, 82, 69, 25, 36, 50, 79, 30, 72, 24, 26, 61, 10, 54, 2, 1] have demonstrated that satellite images, publicly available around the world, can provide important urban semantics related to human activities in urban regions. For example, residential structures identifiable in satellite imagery often indicate regions with high outbound commuting demand, while the presence of commercial buildings typically corresponds to regions with concentrated inbound flows. In contrast, sparsely built-up areas tend to exhibit lower levels of human activity and mobility. Therefore, satellite imagery holds the potential to serve as an alternative to the hard-to-obtain inputs required by traditional computational OD generation models, and supports the generation of OD flows for cities with limited data, as shown in Figure 1.

Accordingly, we propose GlODGen (**Gl**obal-scale **O**rigin-**D**estination Flow **Gen**erator), which incorporates satellite imagery combined with population data as input for representing urban regions, and leverages recently developed vision-language geo-foundation models [84, 42] to extract high-quality semantic features for OD flow generation in any city worldwide. Specifically, satellite imagery is first preprocessed by cropping and stitching it according to the boundaries of each urban region. This step removes irrelevant pixels and avoids the noise outside the regions. Then, with the help of RemoteCLIP [43], the semantic features of urban regions are extracted. Finally, the extracted regional features are combined with population data to profile the urban space as input for the state-of-the-art OD flow generation model, WEDAN [56], and generate OD flows. To investigate the validity of adopting only the publicly available satellite imagery and population data as input, we conduct experiments on representative urban areas in the U.S., Europe, China, Brazil, and Africa. We surprisingly find that totally publicly available data can serve as a perfect alternative to the hard-to-obtain inputs mentioned above and generate OD flows with high consistency with real-world data. This may shed light on the potential for extending OD flow generation to any city worldwide with the help of ubiquitous satellite imagery and population data, which is of great significance for the sustainable development of cities around the world. To facilitate practical use, we release GlODGen as an open-source tool at `https://github.com/tsinghua-fib-lab/generate-od-pubtools`, which seamlessly automates the collection and preprocessing of satellite imagery and population data, urban semantic feature extraction, and OD flow generation together.

The contributions of this paper can be summarized as follows:

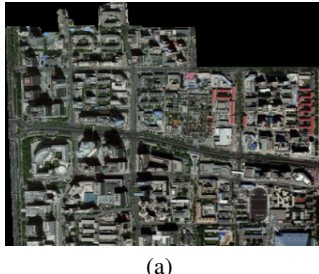 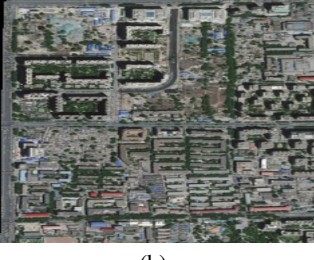 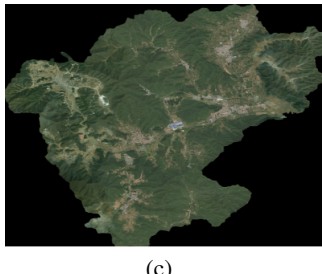

|     (a)     |     (b)     |     (c)     |

Figure 2: Cases of satellite imagery for urban regions.

- We find that publicly available satellite imagery and population data can replace hard-to-obtain urban data, such as sociodemographics, infrastructure data, land use, and POIs, to profile the urban space for inferring human mobility and generate OD flows.

- We propose GlODGen, a novel OD flow generator that leverages vision-language geo-foundation models to extract high-quality semantic features from satellite imagery and generate OD flows via graph diffusion models.

- We implement GlODGen as an efficient tool, which integrates data acquisition, curation and preprocessing, urban semantic feature extraction, and OD flow generation together. With only the boundaries of urban regions given, GlODGen can provide the OD flow data for cities of interest with *only a single line of code*.

- We conduct experiments on representative urban areas around the world to demonstrate the effectiveness of GlODGen, which may pave the way for research on generating OD flows for global cities and contribute to lowering the barrier of data access for sustainable urban development.

## 2 Preliminaries

### 2.1 Definitions and Problem Formulation

**Definition 1. City.** A city refers to a large, integrated urban area such as New York City, London, or Beijing. Our study focuses on commuting movements that occur within the spatial boundaries of a city.

**Definition 2. Urban Region.** Urban regions $\mathcal{R} = \{r_i | i = 1, 2, ..., N\}$ are the basic spatial units within a city for the OD flow generation task. Population movements are modeled between these regions.

*Every region corresponds to its unique satellite imagery*, which consists of 2D, bird's-eye views captured from satellites covering the region, as illustrated in Figure 2.

**Definition 3. Commuting OD Flow.** This refers to the number of people $\mathcal{F}_{r_{org}, r_{dst}}$ who live in one urban region $r_{org}$ and work in another urban region $r_{dst}$. It captures the regular, static daily movement from home to the workplace and remains relatively stable for a long time.

PROBLEM 1. *Commuting OD Flow Generation.* Given a city and its region division, generate the commuting OD flows $\mathcal{F} = \{\mathcal{F}_{r_{org}, r_{dst}} | r_{org}, r_{dst} \in \mathcal{R}\}$ that represent the daily commuting patterns of the city. The generated OD flows should be as consistent as possible with real-world human mobility patterns.

### 2.2 Related Works

**Urban Representation Learning via Satellite Imagery.** Satellite imagery has been demonstrated to carry rich information about human activities, making it a feasible basis for profiling urban regions and generating OD flows. Specifically, satellite imagery offers a comprehensive overview of urban development from a macro perspective, while also enabling detailed analysis of regional contents at a micro level. At a coarse spatial level, satellite imagery has been leveraged to infer socio-economic indicators through tailored supervised and self-supervised learning methods, facilitating

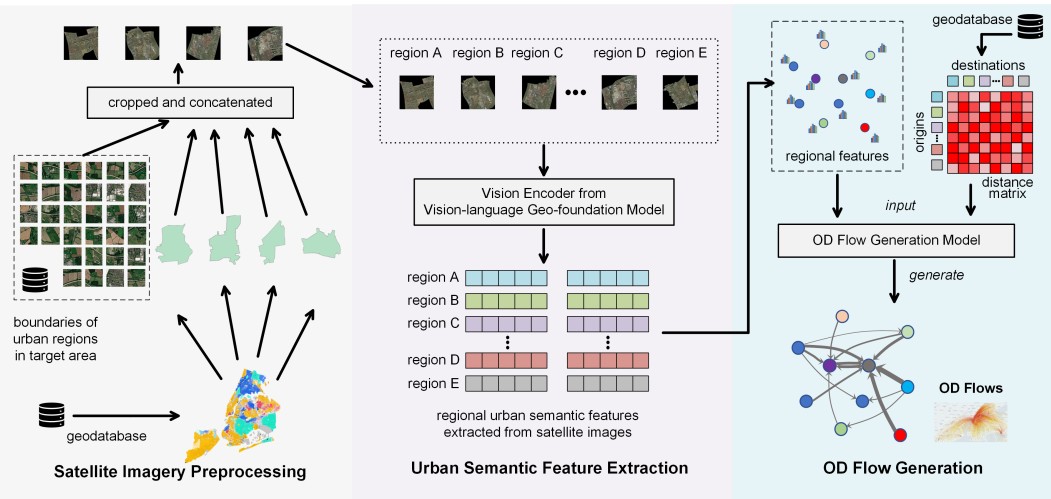

Figure 3: The framework and data pipeline of GlODGen.

the extraction of high-level semantic representations of urban areas. These indicators include poverty levels [5, 4, 25, 36, 50, 79, 73, 27, 41, 82, 24, 61, 10, 2, 1, 72], crop yields [45, 46, 59, 71, 78, 80], land cover [31, 69, 62], commercial activeness [30, 40], and environmental metrics [72, 77, 76, 20, 60, 82, 54, 27]. At a finer spatial scale, satellite imagery has been extensively utilized to monitor urban geospatial features, enabling the identification of various socio-physical entities such as streets, airplanes, vehicles, and recreational facilities like baseball fields [74, 75, 39, 73, 83, 76, 20, 67, 76, 14]. These capabilities collectively highlight the strong potential of satellite imagery as a powerful and scalable modality for comprehensively representing urban regions, thereby enabling the generation of high-quality OD flow data around the world.

**OD Flow Generation.** OD flow obtaining, the task of obtaining OD flows for urban areas of interest lacking human mobility data, is an important yet traditionally costly and time-consuming process that relies on travel surveys [55, 3, 34]. Researchers have explored alternative data sources related to individual trajectories, such as call detail records (CDRs) [11, 34] and cellular network accesss (CNAs) [22, 49] to efficiently access OD flows. However, these methods face challenges regarding data accessibility and privacy concerns. To avoid these issues, researchers have introduced model-based methods for generating OD flows, which are often categorized into two distinct types. The first is physical models, e.g., the gravity model [85] and radiation model [64], which model the population movement by mimicking physical laws, such as Newton's law of universal gravitation and the emission and absorption of the radiation process. These models are overly simplistic and are unable to effectively model the complexity inherent in human mobility. The second category is data-driven computational models that leverage machine learning and deep learning techniques to generate OD flows based on various urban factors, including sociodemographics, POI distributions, etc. [51, 53, 63, 44, 58, 57]. These methods achieve superior performance via sophisticated model structures but the high requirements for input data limits their applicability, especially in underdeveloped areas. Our framework leverages global publicly available data to replace the hard-to-collect features for profiling urban regions and generating OD flows. This breaks the barrier of data accessibility while reserving the superior performance advantages of computational models.

## 3   GlODGen: Commuting Origin-destination Flow Generator

In Figure 3, we illustrate the conceptual structure of the framework. It comprises three stages: *satellite imagery preprocessing*, *urban semantic feature extraction*, and *origin-destination flow generation*.

**Satellite Imagery Preprocessing.** For each region in the city, GlODGen first identifies specific tiles of satellite imagery corresponding to each region and downloads them from any online platform, such as Google Earth and Esri World Imagery. A tile refers to a smaller, square-shaped portion of a larger satellite image, which is created by dividing the satellite image of the whole world into

manageable pieces. This tiling process facilitates more efficient obtaining and processing, especially when working with large-scale satellite images. With all tiles obtained, the next phase involves concatenation and cropping, shaping the images to match the exact contours of each region. This procedure is critical for ensuring that the imagery faithfully represents the geographical and physical attributes of each region, creating an accurate visual portrayal. During cropping, the image content beyond region boundaries is removed and zero-padded to preserve spatial accuracy and exclude irrelevant information, as formalized below:

$$M_r(x, y) = \begin{cases} 1, & \text{if } (x, y) \in \text{Boundary}(r) \\ 0, & \text{otherwise} \end{cases} \quad (1)$$

where $(x, y)$ denotes the pixel coordinates, $M_r$ is the mask of region $r$. The image is then masked by $M_r$ to remove the irrelevant parts, which is detailed in the equation below:

$$I_{\text{final}} = I_{\text{raw}} \odot M_r \quad (2)$$

where $I_{\text{raw}}$ is the raw satellite image, $I_{\text{final}}$ is the final satellite image of region $r$, and $\odot$ denotes the element-wise multiplication operation. Once processed, the satellite images for the regions appear as depicted in Figure 2.

**Urban Semantic Feature Extraction.** After completing the previous step, every region in the specified area is assigned a satellite image to represent its unique characteristics and functions, as illustrated in Figure 3. However, the raw pixel data from the images are unsuitable for direct use as urban regional features, due to the presence of excessive noise and irrelevant, redundant information unrelated to human mobility. As such, a feature extraction model is needed to process the satellite imagery, enabling the distillation of the most significant semantic information related to OD flows. This model could be any structure, such as VGG [65], ResNet [29], or ViT [18], and we choose the vision encoder from the large multimodal model, which also has the special design to handle the satellite images, RemoteCLIP [43]. Pre-trained with natural language supervision for the vision encoder, the large multimodal model exhibits zero-shot capabilities, allowing it to extract versatile semantic features from satellite imagery for various downstream tasks. We adopt RemoteCLIP as the semantic feature extractor in our framework. Specifically, we input the preprocessed satellite image of each region to the vision encoder of RemoteCLIP, whose output is a 1024-dimension high-level feature that captures the urban semantics of that urban region. During this stage, we only use the vision encoder of RemoteCLIP with the pre-trained weights frozen. Because recent works have demonstrated that finetuning the foundation model on the specific task will lead to forgetting the general knowledge learned from the pre-training and result in overfitting [17, 35, 32]. This process is formalized as follows:

$$E_r = \mathcal{V}(I_r), \quad (3)$$

where $I_r$ is the satellite image of region $r$, $\mathcal{V}$ is the vision encoder of RemoteCLIP, and $E_r$ is the 1024-dimension feature vector of region $r$. The final input to the OD flow generation model is built by merging the semantic features, extracted from satellite images, with the population of region $r$, as shown below:

$$X_r = [E_r, P_r], \quad (4)$$

where $P_r$ means the population of $r$ and $[]$ denotes a concatenation operation. $X_r$ is the final features profiling region $r$, which will be input into the OD flow generation model before being processed by a multi-layer perceptron (MLP) whose parameters are trained while training the following OD flow generation model. In this way, the vision encoder can provide a general and versatile representation of the region, while the representation is further refined to fit the specific task of OD flow generation.

**Origin-destination Flow Generation.** We adopt WEDAN [56], a state-of-the-art OD flow generation model based on graph denoising diffusion [38, 47, 70, 23, 7, 21], to generate OD flows. In WEDAN, urban regions are represented as nodes and OD flows as weighted directed edges in a graph. The model follows a conditional generation paradigm, where node attributes serve as guidance for the denoising process and generating the directed edges and corresponding weights. In other words, the semantic characteristics of urban space are leveraged as conditioning inputs, guiding the denoising process to produce realistic and spatially consistent OD flows. Benefiting from its formulation of the city-wide OD network as a directed, weighted graph, WEDAN is capable of generating OD flows that closely align with real-world population mobility patterns. In our framework, the urban semantic feature $X_r$ extracted for each region is used as the node attribute input, guiding the generation of

edge directions and weights during the denoising process. The procedure can be described as follows:

$$p_\theta(\mathbf{F}^{t-1}|\mathbf{F}^t, \mathcal{C}_\mathcal{R}) = \mathcal{N}(\mathbf{F}^{t-1}; \mu_\theta(\mathbf{F}^t, t, \mathcal{C}_\mathcal{R}), (1 - \bar{\alpha}^t)\mathbf{I}), \quad (5)$$

where

$$\mu_\theta(\mathbf{F}^t, t, \mathcal{C}_\mathcal{R}) = \frac{1}{\sqrt{\alpha_t}}(\mathbf{F}^t - \frac{\beta_t}{\sqrt{1 - \bar{\alpha}_t}}\epsilon_\theta(\mathbf{F}^t, t, \mathcal{C}_\mathcal{R})). \quad (6)$$

In the formulas, $\mathcal{C}_\mathcal{R} = \{X_r | r \in \mathcal{R}\}$ is the semantic features of urban regions within the city, $t$ is the diffusion step, $\mathbf{I}$ is the identity matrix, $\epsilon_\theta$ means denoising networks, $\mu_\theta$ denotes $\mu$ of the Gaussian distribution, $\alpha_t = 1 - \beta_t$ and $\bar{\alpha}_t = \prod_{i=1}^t \alpha_i$ refer to the noise scheduler. The denoising network utilized here is the graph transformer network [19], which predicts the noise needed to be removed from the current graph state $\mathbf{F}^t$ to reach the previous state $\mathbf{F}^{t-1}$.

# 4 Experiments

In this section, we conduct experiments to answer two key research questions:

- **RQ1:** For precision, can entirely public data, satellite imagery, and population data, sufficiently represent urban spatial characteristics and support the generation of high-quality OD flows?

- **RQ2:** For generalizability, does GlODGen demonstrate cross-continental transferability, enabling it to generate OD flows in diverse global cities with the help of global public input data?

All experiments were conducted on a single NVIDIA GeForce RTX 4090 GPU (24GB) and an Intel Xeon Platinum 8358 CPU @ 2.60GHz. For all trainable models, we performed the grid search to select optimal hyperparameters. The same training, validation, and testing splits were used when training all models and evaluating the performance.

## 4.1 Performance of Public Satellite Imagery and Population Data (RQ1)

We begin by evaluating the effectiveness of publicly available satellite imagery and population data in supporting OD flow generation. Specifically, we compare the performance of several existing OD flow generation models using two different types of input: (1) traditional fine-grained features that are often costly or difficult to obtain, and (2) publicly accessible features. In this experiment, we use detailed sociodemographic attributes, economic indicators, and POI distributions as the traditional inputs, and satellite imagery combined with population data as the public inputs. To evaluate representational capacity across input types, we apply two groups of existing models commonly used for OD flow generation: physical models, including the classical gravity and radiation models, which rely solely on population distribution act as lower-bound reference of performance; Computational models, including Random Forest [51], DeepGravity [63], GMEL [44], NetGAN [8], and WEDAN [56], which support richer semantic inputs and allow us to assess the performance impact of different input data sources. A detailed introduction of the models is provided in Appendix B.1.

**Dataset.** In this part, we use the dataset [56] from the United States to conduct experiments. This dataset totally consists of 3,333 urban areas, including 3,233 counties and 100 metropolitans. Each urban area is associated with its region division: census tracts within counties and census block groups within metropolitans. The dataset contains two parts: (1) city characteristics in urban regions, including population structure, education level, poverty, income, vehicle ownership, and other socioeconomic indicators from the American Community Survey (ACS) and (2) OD flow data between urban regions provided by the National Census Bureau through the Longitudinal Employer-Household Dynamics Origin-Destination Employment Statistics (LODES) [9].

**Experimental Settings.** To systematically evaluate the performance of different models and input types, we adopt root mean square error (RMSE), normalized RMSE (NRMSE), and common part of commuting (CPC) as metrics. We split the 3,333 urban areas into 8:1:1 proportions for training, validation, and testing. The evaluation metrics are averaged over all test urban areas. All experiments are repeated 5 times with different random seeds with average results and standard deviations reported.

**Experimental Results.** As shown in Table 1, public data demonstrates performance close to that of traditional hard-to-obtain urban data. We can see that the performances of global public data

Table 1: Performance of existing OD flow generation models on test urban areas in the United States with different input data. Traditional hard-to-obtain urban data and global public data are put together for better comparison. The former's results are shown in the upper line, while the latter's are shown in the lower line. Perc. in the table indicates percentage of the performance (CPC) achieved by the model trained with publicly available data relative to that of the model trained with traditional hard-to-collect fine-grained data. Gap means the remaining performance gap in RMSE between the model trained with publicly available data and the model trained with traditional hard-to-collect fine-grained data.

| | CPC↑ | Perc. | RMSE↓ | Gap | NRMSE↓ |
|---|---|---|---|---|---|
| Gravity Model | $0.321 \pm 0.02$ | - | $174.0 \pm 10.4$ | - | 2.222 |
| Radiation Model | $0.347 \pm 0.04$ | - | $196.9 \pm 11.7$ | - | 2.502 |
| Random Forest | $0.494 \pm 0.02$ | - | $100.4 \pm 6.5$ | - | 1.282 |
| | $\mathbf{0.480} \pm 0.03$ | 97.1% | $114.0 \pm 7.1$ | -13.5% | 1.455 |
| DeepGravity | $0.449 \pm 0.01$ | - | $92.9 \pm 9.9$ | - | 1.186 |
| | $\mathbf{0.427} \pm 0.05$ | 95.1% | $99.9 \pm 12.0$ | -7.5% | 1.275 |
| GMEL | $0.462 \pm 0.01$ | - | $94.3 \pm 4.0$ | - | 1.204 |
| | $\mathbf{0.451} \pm 0.02$ | 97.6% | $105.4 \pm 7.9$ | -11.8% | 1.345 |
| NetGAN | $0.517 \pm 0.05$ | - | $89.1 \pm 17.0$ | - | 1.138 |
| | $\mathbf{0.468} \pm 0.06$ | 90.5% | $98.0 \pm 12.3$ | -10.0% | 1.251 |
| WEDAN | $0.634 \pm 0.01$ | - | $64.06 \pm 3.3$ | - | 0.818 |
| | $\mathbf{0.623} \pm 0.02$ | 98.3% | $67.88 \pm 6.1$ | -5.9% | 0.867 |

are slightly less accurate than those of traditional hard-to-collect information, but the deviations are almost indistinguishable. Across all models based on the data-driven schema, public data achieve over 90% of the performance of traditional features. This indicates that such data can further effectively profile and represent spatial characteristics in a manner comparable to traditional hard-to-obtain inputs. Moreover, computational models based on public data significantly outperform classic physical models, demonstrating the superior performance and widespread applicability of public data, including satellite imagery and population data.

## 4.2 Generalization Across Continents (RQ2)

In this part, we conduct experiments that transfer the OD flow generation model from one trained continent to another and investigate the generalization ability of GlODGen with global publicly available satellite imagery and population data. Specifically, we use the data from the United States to train the GlODGen model, and then transfer it to the United Kingdom for evaluation.

**Dataset.** The training data in the United States is the same as the dataset used in Section 4.1. The test data in the United Kingdom is from the Office for National Statistics (ONS) [48], which provides population-level commuting flows between all census units. Specifically, the data contains OD flows of 326 Local Authority Districts (LAD). In each district, the whole area is divided into urban regions by the boundaries of Middle layer Super Output Areas (MSOA).

**Experimental Settings.** We use 90% urban areas in the United States as the training data and the remaining 10% as the validation data to tune the hyperparameters. The trained GlODGen model is then transferred to the United Kingdom for generating OD flows. Generated OD flows will be compared with the ones from census data and evaluated by metrics introduced in Section 4.1. Metrics are the average of all LADs. We also introduce existing models to perform the same task and evaluate their performance as baselines. It is worth noting that existing models are only provided with population data and inter-region distances as input. This limitation is unavoidable, as fine-grained urban features that align across training and testing cities are typically unavailable in cross-continental experimental settings.

**Experimental Results.** As shown in Table 2, the proposed framework can be transferred between different continents with a robust performance. Compared with the baseline models, our framework

Table 2: Performance comparison of baseline methods with geographically distributed population and our proposed framework profiling urban regions based on global public data on the United Kingdom. All the methods are trained in the United States and evaluated in the United Kingdom. IMP. denotes improvement percentage relative to the baselines.

| Model | CPC↑ | IMP. | RMSE↓ | IMP. | NRMSE↓ | IMP. |
|---|---|---|---|---|---|---|
| Gravity Model | $0.240 \pm 0.01$ | -27.5% | $101.6 \pm 3.42$ | +48.9% | $1.752 \pm 0.06$ | +48.9% |
| Radiation Model | $0.323 \pm 0.07$ | -2.4% | $211.6 \pm 4.27$ | -6.4% | $3.647 \pm 0.07$ | -6.4% |
| Random Forest | $0.334 \pm 0.03$ | +0.9% | $223.2 \pm 3.42$ | -12.2% | $3.847 \pm 0.06$ | -12.2% |
| DeepGravity | $0.359 \pm 0.04$ | +8.5% | $157.0 \pm 7.90$ | +21.1% | $2.706 \pm 0.13$ | +21.1% |
| GMEL | $0.362 \pm 0.01$ | +9.4% | $149.1 \pm 10.00$ | +25.0% | $2.570 \pm 0.17$ | +25.1% |
| NetGAN | $0.331 \pm 0.06$ | - | $198.9 \pm 15.21$ | - | $3.429 \pm 0.26$ | - |
| GlODGen | $\mathbf{0.485} \pm 0.03$ | +46.5% | $\mathbf{72.68} \pm 2.13$ | +63.5% | $\mathbf{1.253} \pm 0.04$ | +63.5% |

can achieve a promising performance and outperform baselines on all metrics with a large margin, i.e., 34.0% improvement on CPC, and 28.5% improvement on RMSE and NRMSE. This indicates that 1) public data, i.e., satellite imagery and population data, can effectively be transferred between different continents; 2) the proposed framework can improve the transferability of the computational OD flow generation models; 3) computational models based on public data can beat physical models in cross-continental transfer experiments, supporting a great improvement on global scale OD flow generation. This strongly supports the conclusion that our framework is highly suitable for generating OD flows across diverse urban areas worldwide.

### 4.3 Case Studies on Typical Urban Areas Worldwide (RQ1 & RQ2)

To further validate the effectiveness of the proposed data generator, we collect human mobility-related data from all over the world and apply the OD flow generation to the corresponding urban areas. Specifically, we collect the data from the following typical urban areas and detailed data processing is provided in Appendix B.3.

- **Europe.** We select London and Paris as representative cities in Europe. The OD flow data for London is sourced from ONS [48], while the data for Paris is synthesized by Sebastian et al. [33] using the 2015 French population census provided by the National Institute of Statistics and Economic Studies (INSEE) [68]. Urban regions in Paris are defined at the commune level.

- **China.** We collect OD flow data for Beijing and Shanghai, two major metropolitan centers in China. The Beijing dataset is provided by a leading internet location service provider. The Shanghai data is extracted from Call Detail Records (CDRs) by China's largest telecommunications company, following the method proposed by Iqbal et al. [34]. Urban regions in both cities are defined by subdistricts. All data was collected in 2016.

- **Brazil.** Julio et al. [13] extracted OD flows for the Rio de Janeiro Metropolitan Area (RJMA) in 2014 using CDRs. Urban regions are defined by Municípios.

- **Africa.** We use OD flow data extracted from CDRs collected in Senegal in 2013 as part of the D4D Challenge [16], processed following Iqbal et al. [34]. Urban regions are defined by Arrondissements.

It is worth noting that real-world OD flow data is inherently difficult to collect. Datasets differ in sampling methods, temporal coverage, and spatial granularity, and often contain noise or bias. As such, they do not serve as perfect ground truth, but only as approximate references. In our evaluation, we focus on comparing the spatial patterns and distribution characteristics between generated and observed data, aiming to assess the plausibility and consistency of the generated flows under real-world constraints.

**Experimental Results.** Based on the data collected from the representative urban areas above, we apply the proposed GlODGen framework to generate OD flows and compare the results with the corresponding real-world data. Given the heterogeneous data sources, varying sampling methods, and inherent noise across regions, a unified error-based evaluation metric is not applicable for direct comparison. To quantitatively assess the alignment between the generated and observed OD flows, we

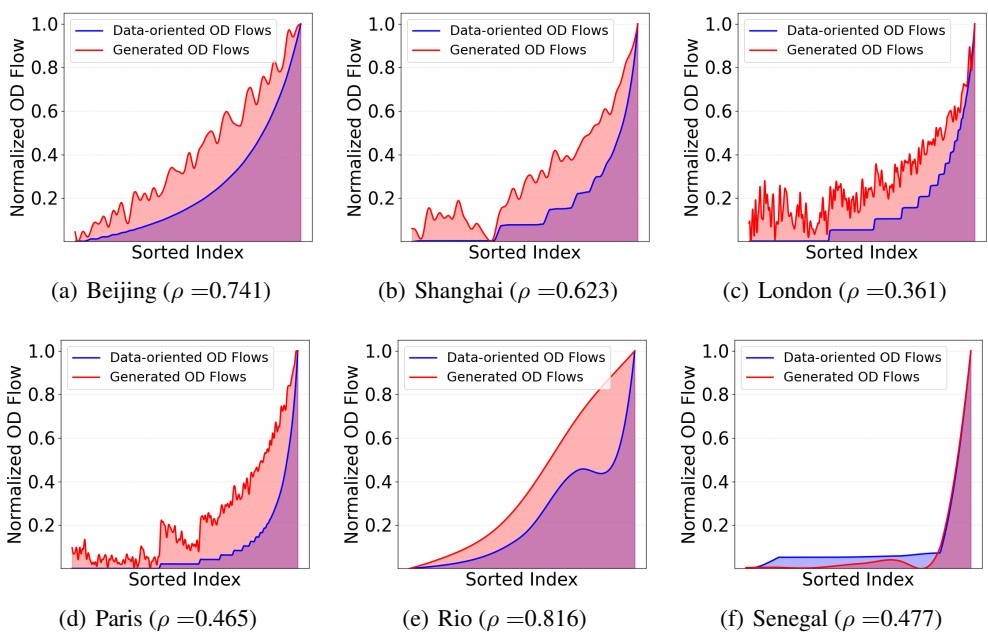

Figure 4: Correlation analysis between generated OD flows and the flows extracted from diverse mobility-related data sources for typical urban areas around the world.

compute the Spearman rank correlation coefficients, visualized as rank-aligned normalized flow curve in Figure 4. Specifically, the data-oriented curve is constructed by the ascending ordered sort of the OD flow oriented from collected data, and the generated flow is also sorted in the same order. Then, the two curves are normalized to the same range and smoothed for better visualization. Although the distribution of the collected data is more skewed, the two curves are aligned with each other in a very similar manner, indicating that the generated OD flows are consistent with the real-world data in terms of data distribution. It is important to note that due to differences in data quality and sources, these correlation values are not directly comparable across cities. In addition, we qualitatively assess spatial patterns by visually comparing the generated and observed OD distributions on a case-by-case basis in Appendix B.4, under varying regional conditions, to provide a more intuitive understanding of the framework's performance. The experimental results show that the OD flows generated by GlODGen closely align with the real-world data in terms of distributional structure, as evidenced by the high Spearman rank correlations observed in most cases. Although lower correlations may arise due to limitations in the reference data itself, high correlation values provide strong empirical support for the credibility and applicability of the proposed framework.

## 5 Discussion

**Conclusion** We find that the global publicly available data, i.e., satellite imagery and population data, has a strong representational power for profiling urban regions and supporting the generation of OD flows across diverse urban areas worldwide. Notably, public data can achieve 98% of the performance of the traditional hard-to-obtain urban data. Building on this insight, we propose a novel data generator, GlODGen, which can generate realistic OD flows for any urban area worldwide. GlODGen integrates vision-language geo-foundation models to extract urban semantic features from satellite imagery and region-level population statistics and generates OD flows through a graph denoising diffusion-based approach. Extensive experiments across all over the world validate both the representational power of public data and the generalizability of GlODGen. For the convenience of practical use, we release GlODGen as an open-source tool, which can automatically complete the data acquisition, curation, preprocessing, urban semantic feature extraction, and OD flow generation with only the boundaries of urban regions given.

**Limitations and Boarder Impact** GlODGen currently has two main limitations. First, the OD flow generation model is primarily trained on U.S. data. While we demonstrate strong cross-continental generalization with the aid of satellite imagery, broader geographic coverage would improve robustness. Second, the generator produces only static commuting OD flows, limiting its applicability to dynamic, fine-grained tasks such as time-dependent traffic analysis. Despite these limitations, GlODGen can benefit various domains, including urban planning, transportation design, energy consumption, carbon emissions, and public health research.

**Ethical Claims** All individual-level location data used in this study have been rigorously anonymized. Additionally, all computations are conducted locally, with no reliance on cloud-based infrastructure, to guarantee both privacy and data security.

### Acknowledgments

This work is supported by the National Natural Science Foundation of China under Grant No. 62476152 and No. U24B20180; and research grants from the Beijing National Research Center for Information Science and Technology (BNRist).

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

# A Additional Details of GlODGen

## A.1 Forward Diffusion Process

We give an introduction to the OD generation model based on graph diffusion in this part. Like the traditional diffusion model, graph diffusion-based models include two main processes: the forward diffusion and the reverse denoising process. During training, the forward diffusion process is utilized to create the training dataset for the denoising network. This process involves gradually adding small Gaussian noise to the original OD flow data, eventually transforming them into pure noise that adheres to a standard Gaussian distribution, as illustrated in the formula below:

$$
q(F_{ij}^t | F_{ij}^{t-1}) = \mathcal{N}(F_{ij}^t; \sqrt{1 - \beta_t} F_{ij}^{t-1}, \beta_t \mathbf{I}),
$$
$$
q(F_{ij}^1, ..., F_{ij}^T | F_{ij}^0) = \prod_{t=1}^{T} q(F_{ij}^t | q^{t-1}),
$$
(7)

where $F_{ij}^t$ denotes the flow starting at $r_i$ flowing into $r_j$ at $t$-th diffusion step, $\mathcal{N}$ is the Gaussian distribution, $\beta_t$ is the noise level at time $t$, and $\mathbf{I}$ is the identity matrix.

The denoising process works in reverse of the forward diffusion process, which is already described in the main paper.

## A.2 Training

The OD flow generation model requires training on diverse datasets to generalize effectively across global scales. We employ data from the whole United States to expose the model to varied mobility patterns, including those from developed and underdeveloped areas.

To generate training samples, the forward process is employed to produce OD flows with varying degrees of noise. Node inputs consist of semantic features extracted from satellite images and population data, while edge inputs are formed by noisy OD flows sampled from the forward process. The model is designed to predict the Gaussian noise disturbances to be removed at each noise level. The training employs the mean squared error (MSE) as the loss function, and optimization is conducted using the Adam optimizer, consistent with [57]. The loss function is formulated as follows:

$$
\mathcal{L} = \mathbb{E}_{t, \epsilon \sim \mathcal{N}(0, \mathbf{I})} \left[ \| \epsilon - \epsilon_\theta(\mathbf{F}^t, t, \mathcal{C}_\mathcal{R}) \|_2^2 \right]
$$
(8)

where $\| \cdot \|$ denotes the $L_2$ norm.

# B Experimental Details

## B.1 Detailed Introduction of Existing OD Flow Generation Models

- **Random Forest. [51]** The Random Forest model is a type of ensemble learning method that builds multiple decision trees and combines their outputs to make a final prediction. It is a popular choice for its simplicity and effectiveness in handling complex relationships between various urban indicators profiling urban regions and OD flows.
- **DeepGravity. [63]** DeepGravity is a multi-layer perceptron (MLP)-based model inspired by the traditional gravity model. It models the process of decision making of destination choice as a classification problem and uses the softmax function to calculate the probability of each destination. With the outflow given, the model can generate the OD flows by multiplying the

probability of each destination and the outflow. In this paper, we directly generate the OD flows by using the model because there is no outflow for any urban areas around the world.

- **GMEL. [44]** GMEL is a graph-based model that uses the graph structure of the urban regions to generate the OD flows. It models the urban regions by aggregating the features of the regions and the features of the connections between the regions. After the geo-contextual embedding learned from the model, random forest is used to generate the OD flows based on the urban region embeddings.

- **NetGAN. [8]** NetGAN is a generative adversarial network (GAN)-based model that minimizes the Wasserstein distance between the random walk sequences of the generated and real OD flows, where the mobility flow networks are represented as graphs. The original NetGAN model is designed for unweighted networks. We adapt it to weighted networks by using the weighted adjacency matrix of the urban regions. The weighted matrix is the OD matrix of the city.

- **WEDAN. [56]** WEDAN is a graph denoising diffusion-based model that uses the graph structure of the urban regions to generate the OD flows. It models the urban regions as nodes and OD flows as the directed weighted edges. The model uses the conditional graph denoising diffusion process to generate the OD flows given the urban region features of the city.

## B.2 Details of Evaluation Metrics

For evaluation, we adopt root mean square error (RMSE), normalized RMSE (NRMSE), and common part of commuting (CPC) as metrics, with the computations detailed below:

$$
\begin{aligned}
\text{RMSE} &= \sqrt{\frac{1}{|\mathbf{F}|} \sum_{r_i, r_j \in \mathcal{R}} ||\mathbf{F}_{ij} - \hat{\mathbf{F}}_{ij}||_2^2}, \\
\text{NRMSE} &= \frac{\text{RMSE}}{\sqrt{\frac{1}{N^2} \sum_{r_i, r_j \in \mathcal{R}} ||F_{ij} - \bar{F}_{ij}||_2^2}}, \\
\text{CPC} &= \frac{2 \sum_{r_i, r_j \in \mathcal{R}} \min(\mathbf{F}_{ij}, \hat{\mathbf{F}}_{ij})}{\sum_{r_i, r_j \in \mathcal{R}} \mathbf{F}_{ij} + \sum_{r_i, r_j \in \mathcal{R}} \hat{\mathbf{F}}_{ij}}
\end{aligned}
\tag{9}
$$

where $\bar{\mathbf{F}}$ represents the expectation of the OD flows $\mathbf{F}$, which is extracted from collected data.

## B.3 Data Processing for Typical Urban Areas Around the World

- **United States** OD flows of urban areas in the United States are collected and provided by the National Census Bureau through the Longitudinal Employer-Household Dynamics Origin-Destination Employment Statistics (LODES) [9]. This dataset provides commuting flows across all census blocks. In line with previous studies [51, 44, 53, 63], we aggregate this data to the census tract level, which defines the regions in our work, while counties represent areas. This dataset offers extensive population coverage and high accuracy, making it widely used in research related to human mobility. In this work, we utilized data collected in 2018.

- **China.** OD flow data are collected for Beijing and Shanghai, two key cities in China. The OD flow data for Beijing is provided by a major internet location service provider in China. Data from Shanghai are extracted from CNAs, provided by China's largest telecommunications company, using the method proposed by Iqbal et al. [34].

- **Brazil.** Julio et al. [13] utilized CDRs to extract OD flows of the Rio de Janeiro Metropolitan Area (RJMA) in 2014. Urban regions in RJMA are defined by Municípios. This dataset provides total flows including repeated records of the same individuals. Therefore, there is a certain degree of bias.

- **Africa.** We identified the dataset of CDRs within Senegal in 2013 [16]. The data was processed to extract OD flows using the method proposed by Iqbal et al. [34]. Urban regions in Senegal are defined by Arrondissements. In Africa, the collection of mobility-related data

is much more challenging due to the lack of comprehensive and up-to-date infrastructure and limited technological resources, and funding of large-scale data gathering efforts. Because of the sparsity and scarcity, the spatial granularity of the data cannot be further refined.

### B.4 Spatial Visualization of Compraison between Generated and Collected OD Flows

We would like to clarify that some visualizations for Beijing, Shanghai, Rio de Janeiro, and a region in Senegal also **appear in a separate submission currently under review at Scientific Data**. In that paper, we present a large-scale dataset of commuting OD flows for 1,625 global cities, generated using the data generator GlODGen introduced in this work, to support research on sustainable urban development.

The core contributions and goals of the two submissions are fundamentally different:

- The Scientific Data submission focuses on the global-scale dataset as a scientific contribution.
- The present paper focuses on the automated generator (GlODGen) and its methodology for worldwide OD flow generation, which highlights the potential of generalization for urban areas around the world.

Although these cities appear in both papers, the visualizations are styled differently and are used for distinct purposes: In Scientific Data, they are presented to validate the quality and scope of the dataset, whereas, in this work, they demonstrate the data generator's generalizability and applicability across diverse global cities. We emphasize that the experimental results are not reused, but rather, the same generated data is used in different contexts to support complementary contributions in the two submissions.

The experimental results of spatial visualization are shown in the following.

To demonstrate the potential of our framework, which relies solely on public data, for generating OD flows globally, we conduct experiments in urban areas including the representative cities in the United States, Europe, China, Brazil, and Africa. Since OD flow data of these diverse areas are collected from different data sources with different sampling biases and noise, we cannot use a unified standard to evaluate the generation of these areas and compare generated OD flows with flows from data in terms of traditional error-based metrics. We visually and qualitatively compare the spatial distribution of the generation and data in a case-by-case manner under different scenarios and data conditions to provide an intuitive understanding of the performance of our framework.

#### B.4.1 China

We generate OD flows for Beijing and Shanghai, two representative cities in China. The visually qualitative comparisons of the generation and data are shown in Figure 5. Figures from above to below are Beijing and Shanghai respectively and the left column is the generation, while the right column is OD flows from data. The generated OD flows exhibit a notable spatial similarity to the data, with the city center and boundary shapes accurately reflected in their spatial distribution. The agreement between generated flows and observed data under the given sampling scheme highlights the framework's validity. Nonetheless, slight discrepancies exist. For instance, flows from Beijing's suburbs to the city center are underrepresented, and generated flows for Shanghai's outskirts are somewhat denser than observed. These differences could stem from the unique urban characteristics of large cities, in which global public data may not fully follow a uniform regularity.

#### B.4.2 Europe

The Greater London and Greater Paris Metropolitan Areas are chosen as representative regions in Europe. The visually qualitative comparisons of the generation and data are shown in Figure 6. Figures from above to below are London and Paris respectively and the left column is the generation, while the right column is OD flows from data. The generated OD flows exhibit substantial similarity to the data. The city boundary is distinctly represented by blue OD flows, while the dense red and yellow lines highlight the city center. Unlike the results observed in China, the generated flows for London successfully capture long-distance OD flows between the city center and the outskirts. In Paris, the generation accurately reflects the limited long trip flows, suggesting that our framework effectively captures Paris's centralized urban structure and efficient planning.

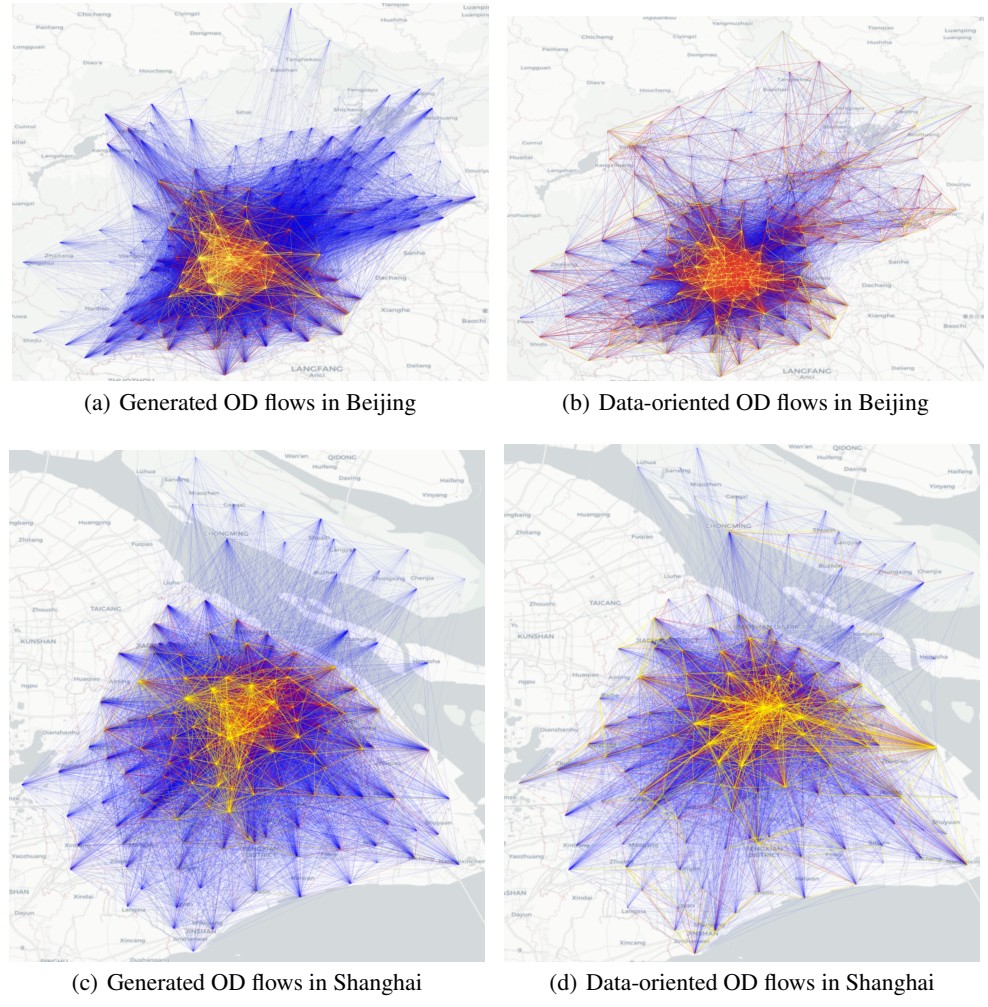

(a) Generated OD flows in Beijing

(b) Data-oriented OD flows in Beijing

(c) Generated OD flows in Shanghai

(d) Data-oriented OD flows in Shanghai

Figure 5: Visualization of the generated and data-oriented OD flows in Beijing and Shanghai. The generated OD flows show high consistency with the data-oriented data in both cities, demonstrating the effectiveness of GlODGen in capturing real-world mobility patterns.

### B.4.3 Brazil

We select the Rio de Janeiro Metropolitan Area as the representative urban area for Brazil. The spatial visualization is presented in Figure 7, demonstrating strong consistency between the generated OD flows and the data. This result highlights the similarity in urban structure and human mobility patterns between Rio de Janeiro and the United States, suggesting that global public data and OD flows from the United States may effectively support OD flow generation for Brazil.

### B.4.4 Africa

Considering the scarcity of data in Africa, we select a representative country in Africa, Senegal, to generate and evaluate our framework. OD flows of Senegal are aggregated from CDRs. The spatial visualization results are presented in Figure 8. As shown, the generated OD flows closely resemble the actual CDR data in terms of spatial distribution. Specifically, the generated OD flow data for Senegal correctly identifies the city center and the urban boundary, which highlights the capability of our framework in accurately modeling mobility patterns even in underdeveloped urban areas.

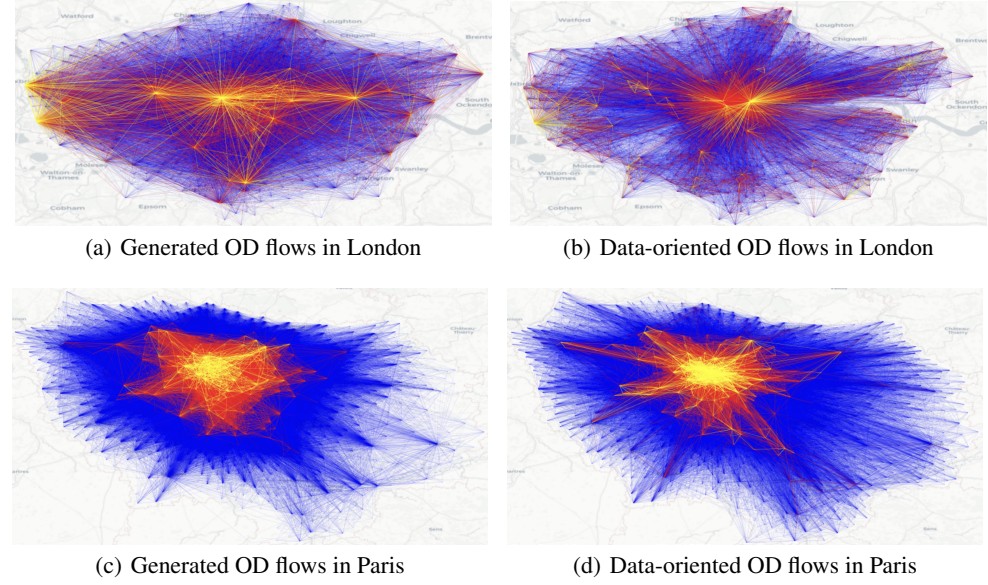

(a) Generated OD flows in London

(b) Data-oriented OD flows in London

(c) Generated OD flows in Paris

(d) Data-oriented OD flows in Paris

Figure 6: Visualization of the generated and data-oriented OD flows in London and Paris. The generated OD flows show high consistency with the data-oriented data in both cities, demonstrating the effectiveness of GlODGen in capturing real-world mobility patterns.

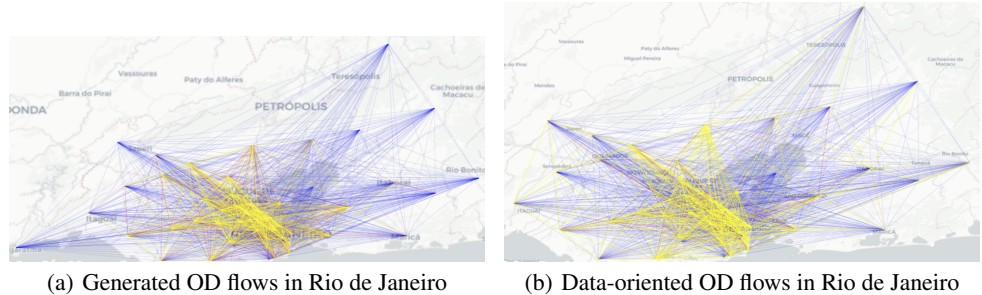

(a) Generated OD flows in Rio de Janeiro

(b) Data-oriented OD flows in Rio de Janeiro

Figure 7: Visualization of the generated and data-oriented OD flows in Rio de Janeiro. The generated OD flows show high consistency with the data-oriented data, demonstrating the effectiveness of GlODGen in capturing real-world mobility patterns.

**B.5**

We conduct a feature importance analysis to investigate the importance of different input features for the OD flow generation. Specifically, we compare the performance degradation of the generation with only one input feature at a time. The results are shown in Table 3. The experimental settings are the same as Table 1. From a theoretical perspective, this is because satellite imagery inherently contains a degree of redundancy related to population information. Previous studies have shown that satellite imagery can be used to estimate population distribution. As a result, using satellite imagery alone can still achieve relatively good performance. However, using population data alone provides an insufficient characterization of urban regions and largely eliminates the representation of regional heterogeneity (e.g., regions with the same population but different functional roles contribute very differently to mobility patterns).

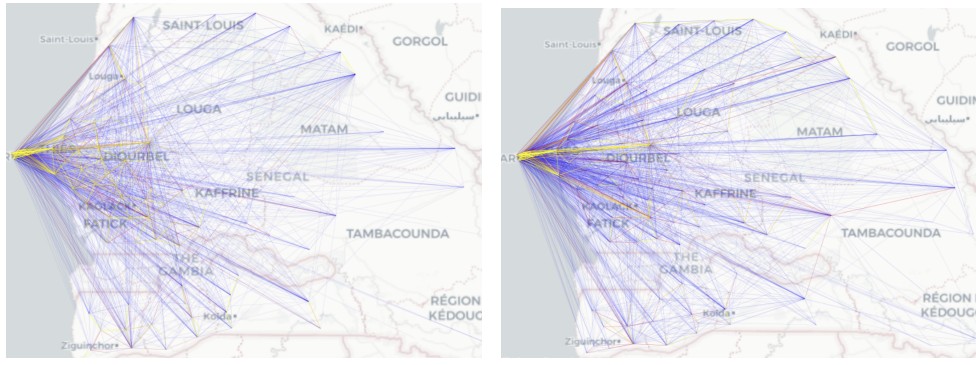

(a) Generated OD flows in Senegal         (b) Data-oriented OD flows in Senegal

Figure 8: Visualization of the generated and data-oriented OD flows in Senegal. The generated OD flows show high consistency with the data-oriented data, demonstrating the effectiveness of GlODGen in capturing real-world mobility patterns.

Table 3: Comparison of the feature importance of population data and satellite imagery for the OD flow generation.

| Input Feature | CPC | RMSE | NRMSE |
|---|---|---|---|
| Population + Satellite | 0.623 | 67.88 | 0.867 |
| Population Only | 0.394 | 137.06 | 1.750 |
| Satellite Only | 0.500 | 92.42 | 1.180 |

