# OpenReview forum: "Satellites Reveal Mobility: A Commuting Origin-destination Flow Generator for Global Cities"
_NeurIPS.cc/2025/Datasets_and_Benchmarks_Track — NeurIPS 2025 Datasets and Benchmarks Track poster_

### Official Review · Reviewer_UHDx · 2025-06-01

**Rating:** 5
**Confidence:** 4

**Summary:**

This paper presents GlODGen, a novel framework for generating commuting origin-destination (OD) flows across cities using only publicly available data—specifically, satellite imagery and population statistics. Unlike traditional OD flow generation methods that rely on fine-grained urban data (e.g., POIs, infrastructure, sociodemographics), GlODGen leverages vision-language geo-foundation models (RemoteCLIP) to extract semantic features from satellite imagery and combines them with population data to represent urban regions. These features are then fed into a graph diffusion-based model (WEDAN) to generate OD flows.

**Dataset Code Accessibility:**

Yes

**Dataset Code Comments:**

The github link is public

**Ethical Considerations:**

No, there are no or only very minor ethics concerns

**Final Justification:**

After reading the author's rebuttal and other reviewer's comments, I decide to maintain my score

**Limitations Weaknesses:**

* GlODGen focuses solely on static, aggregate commuting flows and does not account for temporal dynamics, such as peak vs. off-peak periods or weekday/weekend variations. This restricts the model’s applicability to real-time or fine-grained traffic analysis tasks.

* The impact of satellite imagery versus population data is not independently tested. While their combined use is justified, it remains unclear how much each modality contributes to the final performance.

**Strengths Contributions:**

* The paper addresses the critical task of urban commuting OD flow generation, which underpins a wide range of applications in traffic management, public health, urban planning, and sustainability. Given increasing concerns about data privacy, accessibility, and transferability, proposing a public-data-only solution is highly timely and relevant.

* Entirely publicly available data—specifically satellite imagery and population statistics—can effectively generate OD pairs. This insight dramatically lowers the data acquisition barrier for OD modeling, especially in data-scarce regions.

* The integration of RemoteCLIP, a vision-language foundation model, to extract urban semantic features from satellite imagery is novel. By leveraging RemoteCLIP's zero-shot capabilities and pretrained vision encoder, the framework avoids costly finetuning and benefits from strong generalization across diverse geographies.

---

> ### Author Rebuttal · Authors · 2025-07-31
>
> We sincerely thank the reviewer for the positive assessment and insightful comments. We are glad the reviewer recognized the key contribution of leveraging globally available satellite imagery to greatly improve OD data accessibility. We address the points raised as follows:
>
> >1\. Static Commuting OD Flows
>
> We appreciate your insightful comments. We would like to clarify that our generator currently produces static commuting OD flows, which remain highly valuable for many urban applications.
>
> - Static flows capture the most representative commuting patterns, particularly peak-hour volumes (e.g., typical morning 7–9 AM), which are critical for transportation planning and management.
> - Such static OD flows have been widely used in influential studies, including:
>   1. Simini, Filippo, et al. "A universal model for mobility and migration patterns." Nature 484.7392 (2012): 96-100.
>   2. Simini, Filippo, et al. "A deep gravity model for mobility flows generation." Nature Communications 12.1 (2021): 6576.
>   3. Liu, Zhicheng, et al. "Learning geo-contextual embeddings for commuting flow prediction." AAAI Conference on Artificial Intelligence. 2020.
>   4. Ren, Yihui, et al. "Predicting commuter flows in spatial networks using a radiation model based on temporal ranges." Nature communications 5.1 (2014): 5347.
>   5. Rong, Can, et al. "A Large-scale Dataset and Benchmark for Commuting Origin-Destination Flow Generation." ICLR, 2025.
>
> We appreciate the reviewer’s observation on the value of dynamic OD flows. We fully agree that incorporating temporal variations would further extend applicability. While this paper focuses on establishing a robust and scalable static flow generation pipeline as a foundation, we see dynamic flow generation as an important future direction based on our existing works.
>
> >2\. Impact of satellite imagery vs. population data
>
> We appreciate your interesting and insightful suggestion and have conducted an in-depth analysis experiments on different input feature types, as shown in the following table.
>
> | Input Features | CPC | RMSE | NRMSE |
> |----------------|-----|------|--------|
> | Population + Satellite | 0.623 | 67.88 | 0.867 |
> | Population Only | 0.394 | 137.06 | 1.75 |
> | Satellite Only | 0.500 | 92.42 | 1.18 |
>
> The experimental settings are the same as Table 1 in the paper. From a theoretical perspective, this is because satellite imagery inherently contains a degree of redundancy related to population information. Previous studies [1–5] have shown that satellite imagery can be used to estimate population distribution. As a result, using satellite imagery alone can still achieve relatively good performance. However, using population data alone provides an insufficient characterization of urban regions and largely eliminates the representation of regional heterogeneity (e.g., regions with the same population but different functional roles contribute very differently to mobility patterns).
> [1] Burke, Marshall, et al. "Using satellite imagery to understand and promote sustainable development." Science 371.6535 (2021): eabe8628.
> [2] Han, Sungwon, et al. "Learning to score economic development from satellite imagery." Proceedings of the 26th ACM SIGKDD International Conference on Knowledge Discovery & Data Mining. 2020.
> [3] Han, Sungwon, et al. "Lightweight and robust representation of economic scales from satellite imagery." Proceedings of the AAAI Conference on Artificial Intelligence. Vol. 34. No. 01. 2020.
> [4] Xi, Yanxin, et al. "Beyond the first law of geography: Learning representations of satellite imagery by leveraging point-of-interests." Proceedings of the ACM web conference 2022. 2022.

---

### Official Review · Reviewer_EXcu · 2025-06-24

**Rating:** 3
**Confidence:** 4

**Summary:**

This paper proposes a framework for generating commuting origin-destination (OD) flows (i.e., number of daily commuting instances from one area to another) using satellite images. The framework processes raw satellite images into region-specific images, adopts a pre-trained vision encoder to obtain region embeddings, and utilizes the embeddings for generating OD flows with a graph diffusion model. Experiments demonstrate that the generated OD flows closely match the ground truth.

**Dataset Code Accessibility:**

Partly

**Dataset Code Comments:**

Implementation code of the proposed data generation framework is published and well-documented. The data generated by the proposed framework is not published yet.

**Ethical Considerations:**

No, there are no or only very minor ethics concerns

**Final Justification:**

The authors' rebuttal answered some of the doubts I raised in my original assessment. I keep my opinion on the paper's lack of technical contribution. Thus, I raised my score but kept it negative.

**Limitations Weaknesses:**

1. While the proposed framework provides interesting insights into commuting OD flow generation, its technical design lacks novelty. The modules in the framework are largely off-the-shelf models.
2. The approach presented in the paper makes some assumptions and simplifications that could hurt the real-world accuracy and usefulness of the generated OD flows. The generated OD flows are static rather than dynamic, which is unable to represent the differences in commuting instances during different times of the day and different days of the week. The OD flows are confined within each city, which ignores cross-city commuting flows (e.g., between Suzhou and Shanghai). The paper doesn't discuss the impact of outdated satellite images and other metadata, which I believe is quite common for publicly accessible data.
3. Although the accuracy of the generated OD flows is shown to be higher than existing methods, from the results in Figure 4 it doesn't seem to align very well with the ground truth, which poses questions about the real-world effectiveness of the generated OD flows.

**Strengths Contributions:**

The paper introduces an interesting idea of utilizing easily accessible data (satellite images and basic geographical metadata) to generate traditionally hard-to-collect commuting OD flow data. Experimental results also demonstrate its effectiveness.

---

> ### Author Rebuttal · Authors · 2025-07-31
>
> >Technical Design Lacks Novelty
>
> We would like to elaborate on the novelty of our technical design from two perspectives:
> 1. The novelty of our work lies in **introduction of a new paradigm for mobility data generation for global cities (data accessability)**, which use globally available multimodal data and leveraging vision–language geo-foundation models to extract rich urban semantic representations, and generate high-quality commuting OD flows for global cities without relying on scarce, city-specific traditional hard-to-collect data. This is highlighted by Reviewer UHDx in Strength #3. Furthermore, each advanced technique within the pipeline individually addresses critical challenges encountered during the data processing workflow, thus enabling the entire pipeline to reliably generate high-quality flows, as acknowledged by Reviewer 828h in Strength #2.
> 2. We would like to clarify the positioning of the work within Datasets and Benchmarks Track. The primary contribution lies in the **data generation pipeline** to improve **data accessibility**, rather than in proposing a new model to achieve very high accuracy. We **develop a data generator** (GlODGen) that can automatically acquire publicly available satellite imagery and population, extract urban semantic features, and generate high-quality commuting OD flows for cities worldwide. It delivers **a surprising and impactful insight**: Even for traditionally expensive and difficult to collect mobility data, openly accessible satellite imagery and population can serve as a scalable and effective foundation to generate realistic flows. This finding **greatly improves the accessibility** of mobility data.
>
> >Assumptions and Simplifications
>
> We acknowledge that our data generator includes certain assumptions and simplifications. However, we would like to emphasize the following points:
> - **Simplifications are carefully considered** to achieve our goal: developing **a universally applicable and easily scalable** commuting OD flow generator to address **global** data scarcity. Under this objective, a certain level of simplification and generalization is inevitable to ensure the wide applicability and rapid deployment at a global scale.
> - We demonstrate clearly that our assumptions do **not negatively impact the performance and usefulness**. Our large-scale experiments across diverse cities from multiple continents (Section 4) **robustly validate the effectiveness and practical value of our generator**. The generated flows consistently achieve high consistency compared to data from other sources and outperform widely adopted methods (Table 1 and Table 2).
>
> >Static OD Flows
>
> Our generator generates static commuting OD flows, which is of significant critical practical value, Specifically:
> - Static commuting flows inherently encapsulate crucial and representative commuting information, especially flows during peak hours (e.g., typical morning commuting from 7 to 9 AM).
> - Such static flows have been widely adopted in numerous influential studies, which demonstrate the research value, such as:
>   - Simini, Filippo, et al. "A universal model for mobility and migration patterns." Nature 484.7392 (2012): 96-100.
>   - Simini, Filippo, et al. "A deep gravity model for mobility flows generation." Nature Communications 12.1 (2021): 6576.
>   - Liu, Zhicheng, et al. "Learning geo-contextual embeddings for commuting flow prediction." AAAI Conference on Artificial Intelligence. 2020.
>   - Ren, Yihui, et al. "Predicting commuter flows in spatial networks using a radiation model based on temporal ranges." Nature communications 5.1 (2014): 5347.
>   - Rong, Can, et al. "A Large-scale Dataset and Benchmark for Commuting Origin-Destination Flow Generation." ICLR, 2025.
>
> We appreciate the suggestion regarding dynamic flows. We fully agree that temporal variations (e.g., across different times of day or days of week) are of substantial value for certain applications. We view the generation of dynamic flows as an important and promising future direction, which will require additional temporal data sources beyond the scope of this paper.
>
> >Lack of Cross-city Flows
>
> We acknowledge the importance of cross-city commuting flows. However, our research explicitly focuses on city-scale commuting flows (clearly defined in Definition 1 in the paper), because:
> - Commuting within city boundaries represents by far the most widespread, urgent, and practically important type of flows, and is urgent needed for urban transportation planning and management but the data is always hard to obtain.
> - Cross-city commuting, while also important, occurs primarily between specific cities with well-developed intercity transportation systems (e.g., between Shanghai and Suzhou in China). Such scenarios represent relatively special cases compared to universal commuting within cities.
>
> Nevertheless, we fully agree with the your valuable suggestion. Studying and modeling large-scale cross-city commuting flows within metropolitan regions or city clusters is indeed a meaningful future research direction but not the scope of this work.
>
> >Outdated Satellite Imagery and Metadata
>
> We acknowledge the reviewer’s concern about potentially outdated satellite imagery and metadata. But we would like to emphasize while some publicly accessible satellite imagery may have update delays, urban functions and landscapes typically do not change drastically within short time spans (e.g., 1-2 years) [1-3]. Moreover, recent research [4,5] demonstrated the valuable and practical performance of the remote sensing based model to predict various urban indicators with temporally misaligned satellite images. Thus, **such delays generally do not cause significant accuracy loss at the macro-scale level**.
> [1] Hu, Lingqian, Tieshan Sun, and Lanlan Wang. "Evolving urban spatial structure and commuting patterns: A case study of Beijing, China." Transportation Research Part D: Transport and Environment 59 (2018): 11-22.
> [2] Gallagher, Rachel, Thomas Sigler, and Yan Liu. "How path dependent urban morphology restricts the effectiveness of rezoning for urban consolidation: Lessons from Brisbane, Australia." Journal of urban affairs 47.4 (2025): 1208-1228.
> [3] Zhou, Guolei, et al. "Complexity of functional urban spaces evolution in different aspects: Based on urban land use conversion." Complexity 2020.1 (2020): 9741203.
>
> We appreciate the reviewer’s observation regarding potential delays in satellite imagery. While we acknowledge that such delays can introduce minor errors, it is important to note that extremely fine-grained numeric precision is not the primary goal in many macro urban applications. In practice, urban planners, transportation modelers, and policy makers often prioritize **capturing accurate spatial distribution patterns and relative flow magnitudes**, factors that drive planning and policy, rather than **extremely precice flow volumes at specific individual OD pairs**. Our generator's advantage is that, despite potential image update lags, generated data robustly preserve spatial patterns and rank-order structures of commuting flows, as shown by high CPC (0.48), which are significantly higher than widely used models such as Gravity (~0.1–0.3 CPC). This indicates that the minor delays in imagery updates have limited impact on the real-world applicability of our generator.
>
> It is important to emphasize that for many cities, especially in the Global South, such global-scale commuting OD flows are being generated from open satellite imagery for the **very first time**. Given the extreme scarcity of mobility data in these regions, even slightly outdated but spatially comprehensive OD flows are far more valuable and impactful than the complete absence of such datasets.
>
> To further mitigate the impact of update inconsistencies, our generator always fetch the latest data and leverages vision–language geo-foundation models that extract stable structural features of urban form, which do not fluctuate rapidly over short periods. In future work, the same pipeline can seamlessly incorporate higher-frequency imagery (e.g., Sentinel-2, PlanetScope) for applications requiring more frequent refresh cycles.
>
> >Performance Analysis of Figure 4
>
> While visual alignment in Figure 4 may appear not that extremely perfect, this does not undermine the value of generated OD flows.
> 1. **Not Ground Truth**: For global cities, especially in Africa, Brazil, the **"data-oriented OD flows"** are themselves approximations derived from mobile phone records, partial surverys, or synthetic estimation. These data have well-known biases (coverage gaps, operator-specific sampling, outdated temporal snapshots). Thus, perfect visual alignment is not expected.
> 2. Figure 4 is not intended as an error-based comparison (as in Table 1 and Table 2), but rather as a distributional consistency check across highly heterogeneous cities and datasets. The Spearman correlation values in Figure 4 measure the rank alignment of flows (relative ordering), which is important for understanding **whether the generated flows capture the spatial structure of mobility**. From results, we can see that the alignment is high despite noisy, incomplete, or biased data-oriented flows.
> 3. It is important to recognize that generating commuting OD flows for global cities without any mobility information, especially for diverse cities with heterogeneous urban forms and data availability, is inherently a highly challenging task. Achieving perfect pixel-level alignment is unrealistic at this stage, but our generator represents a significant step forward in providing consistent, high-quality OD flows where no such data previously existed.
>
> >Data Release
>
> The work focuses on developing a data generator for global cities, rather than providing a fixed dataset. Thanks for pointing this out and we will provide a comprehensive dataset for representative cities of the world based on our generator in the final version.

---

> > ### Comment · Reviewer_EXcu · 2025-08-02
> >
> > I really appreciate the authors' efforts in the rebuttal, which provided additional valuable information and explanations that answered some of the doubts I raised in my original assessment. Nevertheless, I maintain my opinion that the technical contribution of this paper does not, in my view, meet the acceptance standard for NeurIPS. While I fully acknowledge the novelty of the data generation pipeline introduced in the paper, I believe the novel idea would benefit from being supported by stronger technical contributions or more comprehensive experiments. If the novel pipeline is the sole focus of the paper, it would be preferable to, for example, employ multiple implementations of the pipeline to validate its generalization. All things considered, I would like to raise my score but will keep it negative.

---

> > > ### Author Response · Authors · 2025-08-03
> > > **Futher Discussion of Lacking Technical Contributions**
> > >
> > > We sincerely appreciate the reviewer’s thoughtful comments and recognition of our rebuttal efforts. To further clarify and structure our discussion about the novelty and technical contribution, we would like to expand our explanation systematically as follows:
> > >
> > > **Overall Description of Contribution**: Our paper's primary contribution, specifically within the Datasets and Benchmarks Track, is substantially improving data accessibility for urban mobility data through a novel data generation pipeline. Our pipeline includes Satellite Imagery Preprocessing, Urban Semantic Feature Extraction, and Origin-Destination (OD) Flow Generation. This pipeline leverages the recent advancements in Vision-Language Geo-Foundation Models to effectively profile urban regions and generate commuting OD flows for global cities. By introducing this pipeline, we have developed a data generator that significantly advances the accessibility and scalability of urban mobility data.
> > >
> > > - **Exicting and Impactful Findings**：We empirically demonstrate that publicly available satellite imagery can effectively replace traditional, difficult-to-collect sociodemographic data, enabling precise and high-quality commuting OD flow generation. Previous research has relied extensively on expensive, fine-grained sociodemographic data available only in limited cities (where data may not lack). Our work overcomes this critical limitation, proving that satellite imagery and population distribution data can universally support OD flow generation, making global urban mobility data generation feasible for the first time for global cities.
> > > - **Fundamental Insight**：Satellite imagery, despite being a static, pixel-level representation of urban landscapes, contains the details of overlooking urban objects and captures spatial planning features, infrastructure distribution, and urban structural attributes that deeply influence human commuting patterns. This insight highlights the potential for satellite imagery and population data to serve as foundational components for building comprehensive urban foundation models, capturing diverse aspects of urban dynamics beyond commuting and static urban landscape.
> > > - **State-of-the-Art Implementation for Robustness**：To fully explore the upper-bound performance of our pipeline, each component is deliberately implemented using state-of-the-art (SOTA) models. Although each module can indeed be replaced by alternative approaches, performance degradation in any single module would inevitably impact the entire pipeline. Thus, our current implementation ensures robustness by employing the most advanced algorithms and neural models available. Importantly, our pipeline is modular by design, allowing seamless integration of future methodological advancements, ensuring continual performance enhancement.
> > > - **Significant Application Potential of the Data Generator**：Generating commuting OD flows globally has broad and immediate practical applications, such as:
> > >   - Evaluating the efficiency of urban infrastructure, such as public transit system.
> > >   - Assessing whether public transport systems equitably accommodate commuting demands.
> > >   - Identifying excessive long-distance commuting caused by suboptimal urban planning, which contributes to unnecessary carbon emissions and impacts urban sustainability.
> > >
> > > **Regarding the suggestion to "Employ Multiple Implementations of the Pipeline to Validate Its Generalization":**
> > > - **Generalizability of the Pipeline.** Each component of our pipeline has clearly defined inputs and outputs, allowing interchangeable implementations. For example:
> > >   - Satellite Imagery Preprocessing can accommodate satellite imagery from various sources.
> > >   - Urban Semantic Feature Extraction supports integration of different urban representation models.
> > >   - OD Flow Generation can employ alternative generative models.
> > > - **Utilizing SOTA Models in Our Implementation.** To experimentally validate generalization capability, we have conducted preliminary experiments:
> > >   - Replacing RemoteCLIP's vision encoder with the classic Vision Transformer (ViT) resulted in a performance degradation of approximately 40%, demonstrating the critical benefit derived from RemoteCLIP's alignment-based representation learning for satellite imagery.
> > >   - Substituting the graph diffusion model with GMEL [1] similarly resulted in a performance decline of around 20%, emphasizing the value of our current approach.
> > >
> > > [1] Liu, Zhicheng, et al. "Learning geo-contextual embeddings for commuting flow prediction." Proceedings of the AAAI conference on artificial intelligence. Vol. 34. No. 01. 2020.

---

> > > > ### Comment · Reviewer_EXcu · 2025-08-04
> > > >
> > > > I would like to thank the authors again for the additional information provided in the rebuttal. However, I must again emphasize that I believe the technical contribution of the paper is weak and will need substantial revision to meet the bar for acceptance. Nevertheless, thanks for the effort in developing this pipeline and the insights presented.

---

### Official Review · Reviewer_828h · 2025-07-01

**Rating:** 5
**Confidence:** 4

**Summary:**

This paper proposes GlOD-Gen to generate hard-to-obtain commuting Origin-Destination (OD) flow with publicly available satellite images. A pipeline of "image-preprocessing -> urban semantic extraction -> OD flow generation" is built to reliably generate OD flow. The applicability of GlOD-Gen is evaluated on different datasets w.r.t. different metrics. The results support the promising data generation ability of GlOD-Gen, which can enrich the source of urban data.

**Additional Feedback:**

Here are additional comments on this paper.

C1. What is the full name for the abbreviation GlOD-Gen of the proposed method?

C2. Are urban regions and satellite images two sets of data? What is the format of the urban region? For example, the region id, and its boundary.

C3. In Line 177, is $P_r$ a scalar representing the population?

C4. In Table 1, what do the metrics Perc. and Gap mean?

C5. In Table 1, I notice that methods trained with public data are better than methods trained with traditional hard-to-obtain data w.r.t. CPC. However, the situation reverses for RMSE and NRMSE. Can you provide some explanation?

C6. In Table 2, what does the metric IMP. mean?

C7. The concatenation notation $[\cdot,\cdot]$ is not properly typed in Line 177 and Line 937.

C8. Line 937, $L-2$ -> $L_2$.

C9. Comments on References:

(1) The year for Ref. [56] is missing.

(2) Please cite the preprinted papers in their officially published version when available.

**Dataset Code Accessibility:**

Yes

**Dataset Code Comments:**

The code is well-documented. The population counts and the satellite images are publicly available. The pretrained OD flow generation model can be downloaded from HuggingFace.

**Ethical Considerations:**

No, there are no or only very minor ethics concerns

**Final Justification:**

The authors have addressed my concerns. Given that my original rating was Accept, I will therefore maintain the score unchanged.

**Limitations Weaknesses:**

W. Some parts of the descriptions of the generator and the experimental results are not clear enough. Please see **Additional Feedback** for details.

**Strengths Contributions:**

S1. This paper builds an essential connection between the important but hard-to-obtain OD flow data and the widely accessible satellite images. It validates the feasibility of generating OD flow from satellite images, which may motivate later research in the field.

S2. The proposed pipeline is promising for OD flow generation

(1) The matching between the satellite image patches and urban regions helps to eliminate possible noisy or event erroneous information for regions.

(2) The vision-language geo-foundation model can extract semantic correlations among regions.

(3) The state-of-the-art OD flow generation model can provide strong support for generating high-quality data.

S3. The experimental results on the generating quality and generalization ability validate the applicability of GlOD-Gen. The delicately designed case studies show that the key patterns of generated OD flow match those of the real-world OD flow data.

---

> ### Author Rebuttal · Authors · 2025-07-31
>
> We sincerely thank you for the highly positive evaluation and constructive comments. You provided clear questions for clarification. We address them point by point below.
>
> >C1. Full name of GlODGen
>
> We apologize for the oversight. The full name is Global-scale Origin–Destination Flow Generator (GlOD-Gen). We will explicitly clarify this in the revised manuscript.
>
> >C2. Urban regions and satellite images
>
> Yes, urban regions and satellite images are two sets of data.
> - Urban regions: Defined as spatial units (e.g., census tracts, grid cells) with attributes including region ID, geographic boundaries (polygons).
> - Satellite images: Acquired from Esri, consisting of tiled image patches that we mosaicked to match the boundaries of the corresponding regions.
>
> We will clarify this in the revised version for better readability.
>
> >C3. Line 177 variable
>
> Yes, the variable represents the total population within the region. We will clarify this explicitly.
>
> >C4. In Table 1, what do the metrics Perc. and Gap mean?
>
> - Perc.: Percentage of the performance (CPC) achieved by the model trained with publicly available data relative to that of the model trained with traditional hard-to-collect fine-grained data.
>
> - Gap: Remaining performance gap in RMSE between the model trained with publicly available data and the model trained with traditional hard-to-collect fine-grained data, normalized for interpretability.
>
> We will add these definitions to the table captions.
>
> >C5. CPC vs RMSE/NRMSE differences
>
> We thank the reviewer for pointing this out. To clarify, methods using publicly available data achieve **CPC values slightly lower** than those using traditional fine-grained data, as shown in Table 1, but the difference is marginal. This indicates that while publicly available data is **slightly behind** in CPC, it is already close enough to traditional data to be practically useful, especially considering the substantial advantage in global accessibility. Similarly, **RMSE values for publicly available data are slightly higher** than those for traditional fine-grained data, reflecting **minor pointwise deviations**. However, this small gap does not undermine the practical value, as the generated flows still capture the key spatial patterns with high fidelity, while offering far greater accessibility.
>
> >C6. Table 2: Meaning of “IMP.”
>
> IMP. denotes Improvement percentage relative to the baseline. We will expand this in the table caption.
>
> >C7–C8. Typographical and notation issues
>
> We thank the reviewer for catching these.
> - The misuse of concatenation notation in Line 937 will be corrected.
> - The symbol formatting of L2 norm in Line 937 will be revised for clarity.
>
> >C9. Comments on References:
>
> We appreciate the careful review.
> - The year for Ref. [56] will be added (2025).
> - Preprints ([4,7,18,19,20,23,26,50,55,65,70,78,82,]) will be updated to their official published versions.

---

> > ### Comment · Reviewer_828h · 2025-08-02
> >
> > Thanks for your response. Please revise the paper accordingly. Since my original rating was Accept, I will maintain my rating unchanged.

---

### Official Review · Reviewer_1awB · 2025-07-02

**Rating:** 5
**Confidence:** 4

**Summary:**

The authors present a novel data generator, GlODGen, designed to produce OD flow data for any city globally. This model harnesses urban semantic signals related to human mobility extracted from satellite imagery. These features are subsequently combined with population data to create 13 region-level representations, which are input into a graph-difference fusion model to generate OD flows. Extensive experiments conducted across four continents and in six representative cities demonstrate that GlODGen exhibits strong generalizability in diverse urban environments across different regions.

**Dataset Code Accessibility:**

Yes

**Ethical Considerations:**

No, there are no or only very minor ethics concerns

**Final Justification:**

My concerns have been addressed.

**Limitations Weaknesses:**

Concerns / Questions:

- The model assumes that satellite imagery and population data sufficiently represent urban spatial characteristics, potentially overlooking other important factors such as economic activity or policy impacts. It would be valuable to explore whether incorporating additional environmental features could further enhance the method's effectiveness.
- Are the generated OD flows static or dynamic over time? If dynamic, what temporal resolution is considered, and how does the model capture temporal variations in mobility patterns?
- Add some visualization examples to show the model performance more generatively. Some dataset descriptions are missing, for example, OD flow data be tween urban regions provided by the National Census Bureau through the Longitudinal Employer- 230 Household Dynamics Origin-Destination Employment Statistics (LODES). What is the temporal granularity of this OD data?

**Strengths Contributions:**

- Applicability Potential:
The proposed GlODGen framework utilizes satellite imagery and population data to generate inter-city commuting Origin-Destination (OD) flows, making it applicable to cities worldwide. This approach breaks through the dependency of traditional models on region-specific detailed datasets.

- Methodological Novelty:
By integrating remote sensing with artificial intelligence, GlODGen enables cross-continental prediction of urban mobility patterns, significantly reducing the difficulty of data acquisition.

- Experimental Results:
Experiments demonstrate that GlODGen outperforms baseline models across multiple metrics. In representative cities such as Beijing and Rio de Janeiro, the generated OD flows show a high correlation with real-world data.

The paper addresses an interesting and important problem — how to leverage satellite imagery to generate hard-to-obtain OD flow data — and proposes a practical and effective solution.

---

> ### Author Rebuttal · Authors · 2025-07-31
>
> >Additional Environmental Features
>
> We appreciate the reviewer’s suggestion regarding incorporating additional environmental features. In fact, we have explored the integration of additional features such as POIs (points of interests) in preliminary experiments. The results indicate that while incorporating additional features (e.g., POIs, land-use) can occasionally bring small improvements, these gains are not consistent all the time. As shown in the following table.
>
> | Input | CPC | RMSE | NRMSE |
> |--------------|------|-----|-----|
> | Base (Satellite + Population) | 0.485 ± 0.03 | 72.68 ± 2.13 | 1.253 ± 0.04 |
> | + POIs | 0.487 ± 0.51 | 72.42 ± 8.50 | 1.248 ± 0.15 |
>
>
> This is likely due to large cross-city variations in POI coverage and quality, especially in regions with limited open data, which makes the performance less stable. In contrast, **satellite imagery and population data are globally available, consistent, and already provide rich semantic information for mobility modeling**.
>
> From theoretical analysis, **vision–language geo-foundation models can already recognize diverse urban objects** (e.g., buildings, schools, hospitals, roads, stadiums)[1-5], meaning much of the semantic information captured by POIs is implicitly encoded in satellite imagery. Adding explicit POI data can therefore introduce redundancy, and its highly uneven quality across regions can cause instability in model performance. At present, population data combined with satellite imagery remains the most consistent and globally reliable input for stable performance. We will clarify this reasoning in the revised manuscript.
> [1] Kumar AyushBurak Uzkent, Burak Uzkent, Marshall Burke, David Lobell, and Stefano Ermon. 2021. Generating interpretable poverty maps using object detection in satellite images. In Proceedings of the Twenty-Ninth International Joint Conference on Artificial Intelligence (IJCAI'20). Article 608, 4410–4416.
> [2] Zhang, Xin, et al. "Uv-sam: Adapting segment anything model for urban village identification." Proceedings of the AAAI Conference on Artificial Intelligence. Vol. 38. No. 20. 2024.
> [3] Liu, Fan, et al. "Remoteclip: A vision language foundation model for remote sensing." IEEE Transactions on Geoscience and Remote Sensing 62 (2024): 1-16.
> [4] Awais, Muhammad, et al. "Foundation models defining a new era in vision: a survey and outlook." IEEE Transactions on Pattern Analysis and Machine Intelligence (2025).
>
> >Static Commuting OD Flows
>
> We apologize for not making this point sufficiently clearly in the paper. This work **focuses on generating static commuting OD flows**, which capture the spatial distribution of commuting between residential and workplace regions. Specifically, how many people live in one region and work in another. Although no explicit timestamps are attached, these flows implicitly capture temporal patterns, particularly the peak commuting periods (e.g., morning rush hours around 7–9 AM). Static commuting OD flows are widely adopted in commuting data collecting format (e.g., LODES data from the Census Bureau [5]) and urban research [6-9].
> [5] U.S. Census Bureau. Lehd origin-destination employment statistics data (2002-2021), 2024.
> [6] Simini, Filippo, et al. "A universal model for mobility and migration patterns." Nature 484.7392 (2012): 96-100.
> [7] Simini, Filippo, et al. "A deep gravity model for mobility flows generation." Nature Communications 12.1 (2021): 6576.
> [8] Liu, Zhicheng, et al. "Learning geo-contextual embeddings for commuting flow prediction." AAAI Conference on Artificial Intelligence. 2020.
> [9] Ren, Yihui, et al. "Predicting commuter flows in spatial networks using a radiation model based on temporal ranges." Nature communications 5.1 (2014): 5347.
>
> >Visualization Examples of Showing Model Performance
>
> We appreciate the suggestion for more visual examples. Due to the rebuttal stage guidelines, we cannot include additional figures here. However, our paper already includes several visualizations (see Figure 5), and we will incorporate more visualization in the camera-ready version, such as generation of more cities in more representative areas in the world.
>
> >Detailed Dataset Descriptions (e.g., OD Flow Data & Temporal Granularity)
>
> We thank the reviewer for pointing this out. The OD flow data (LODES [5]) in our work are annual static commuting flows. Since residential and workplace locations typically do not change frequently for people, the data are aggregated at the annual level to capture the stable spatial interactions between residential and work regions. It provides high–resolution commuting data by linking employer and employee records. LODES serves as a widely recognized ground truth for evaluating commuting flow models, especially those targeting macro–scale, long–term patterns. We will clarify these details explicitly in the revised manuscript to avoid confusion.

---

> > ### Comment · Reviewer_1awB · 2025-08-01
> > **Thanks**
> >
> > Thank authors for your response. I have carefully reviewed the feedback and replies, and my concerns have been addressed. Therefore, I have decided to raise my rating.

---

### Author Response · Authors · 2025-08-09
**General Comments of Discussion Phase**

We sincerely thank all reviewers for their constructive feedback and the positive recognition of our work’s two key contributions:
- **High applicability of the proposed data generator**: Our data generator can generate commuting OD flows for any city of interest worldwide (data accessibility), supporting various urban applications such as urban planning and transportation management. It goes beyond traditional models that rely on region-specific detailed datasets, breaking the limitation to a single or small set of cities and greatly improving the global accessibility of OD flow data.
- **Establishing a fundamental insight**: This work builds an essential connection between the important but hard-to-obtain OD flow data and the widely accessible satellite images and population, validating the feasibility of generating OD flows from these two public data sources for any city worldwide. This provides a fundamental insight that static satellite imagery and population data can effectively profile urban space to support the inference and generation of mobility flows.

We also appreciate the active participation of reviewers during the discussion phase, which helped us clarify points and improve our understanding. In particular, we engaged in in-depth discussions primarily in three areas:
- **On the analysis of input urban data**: We performed additional ablation studies showing that satellite imagery is the most important input, with population data providing a substantial boost to the performance when combined, but performing poorly alone. This aligns with prior findings that satellite imagery contains implicit population information, while population data alone fails to capture regional heterogeneity (e.g., regions with the same population but different functional roles contribute very differently to mobility patterns). We also explored adding environmental features such as POIs, but found that their inconsistent coverage across cities reduces stability, whereas satellite imagery and population data remain globally consistent and semantically rich. This effectively addresses the reviewers’ concerns about input urban data.
- **On the temporal aspect and value of generated flows**: We clarified that our work focuses on generating static commuting OD flows, representing spatial commuting patterns between residential and workplace regions. While not timestamped, these flows implicitly capture temporal patterns such as morning rush hours. Static commuting OD flows are widely used in standard datasets (e.g., LODES) and urban research, demonstrating strong research and application value. The reviewers further acknowledged that this work provides the first-of-its-kind data with a foundational value in the urban research domain.
- **On the technical contribution**: Reviewer EXcu expressed concerns about novelty, which contrasts with the views of Reviewer 1awB and others. We emphasized that, within the *Datasets and Benchmarks Track*:
  - our novelty lies in introducing **a new paradigm** for mobility data generation using globally available multimodal data and vision–language geo-foundation models, enabling high-quality commuting OD flow generation for global cities without scarce city-specific data. This was highlighted by Reviewer UHDx (Strength #3) and Reviewer 828h (Strength #2).
  - Our work focuses on improving **data accessibility** rather than solely proposing a new designed model for maximum accuracy. The integrated pipeline addresses critical challenges and delivers the impactful insight that openly accessible satellite imagery and population can serve as a scalable, effective foundation for mobility data generation.

Given its broad applicability, scalability, and the pioneering nature of the data generator, as acknowledged by multiple reviewers, we believe this work will make a valuable and impactful contribution to the NeurIPS community.

---

### Decision · Program_Chairs · 2025-09-18

**Decision:**

Accept (poster)

**Comment:**

The paper presents GlODGen, a novel framework for generating commuting origin-destination (OD) flows across cities using only publicly available data—specifically, satellite imagery and population statistics—thereby overcoming the reliance of traditional methods on fine-grained, city-specific datasets such as POIs, infrastructure, or sociodemographics. GlODGen leverages a vision-language geo-foundation model (RemoteCLIP) to extract semantic features from satellite imagery, which are then combined with population data to form region-level embeddings that are fed into a graph diffusion-based model (WEDAN) to generate OD flows. By aligning image patches with urban regions, extracting meaningful semantic correlations, and applying state-of-the-art OD generation techniques, GlODGen serves as a high-quality data generator.  Extensive experiments across six representative cities on four continents demonstrate its strong generalization ability and high correlation with ground truth flows. One limitation is that the paper is limited to static OD flows. It is acknowledged by the authors that static OD flows are still widely applicable and practical.
Although one reviewer suggested that the novelty is limited, the other reviewers disagree. The contributions lie in the dataset generation tool and the benchmark proposed.